# Endogenous Neuropeptide Nocistatin Is a Direct Agonist of Acid-Sensing Ion Channels (ASIC1, ASIC2 and ASIC3)

**DOI:** 10.3390/biom9090401

**Published:** 2019-08-22

**Authors:** Dmitry I. Osmakov, Sergey G. Koshelev, Igor A. Ivanov, Yaroslav A. Andreev, Sergey A. Kozlov

**Affiliations:** 1Shemyakin-Ovchinnikov Institute of Bioorganic Chemistry, Russian Academy of Sciences, 117997 Moscow, Russia; 2Institute of Molecular Medicine, Sechenov First Moscow State Medical University, 119991 Moscow, Russia

**Keywords:** acid-sensing ion channel (ASIC), endogenous neuropeptide, electrophysiology, signaling, neuroscience, agonist of receptor

## Abstract

Acid-sensing ion channel (ASIC) channels belong to the family of ligand-gated ion channels known as acid-sensing (proton-gated) ion channels. Only a few activators of ASICs are known. These are exogenous and endogenous molecules that cause a persistent, slowly desensitized current, different from an acid-induced current. Here we describe a novel endogenous agonist of ASICs—peptide nocistatin produced by neuronal cells and neutrophils as a part of prepronociceptin precursor protein. The rat nocistatin evoked currents in *X. laevis* oocytes expressing rat ASIC1a, ASIC1b, ASIC2a, and ASIC3 that were very similar in kinetic parameters to the proton-gated response. Detailed characterization of nocistatin action on rASIC1a revealed a proton-like dose-dependence of activation, which was accompanied by a dose-dependent decrease in the sensitivity of the channel to the protons. The toxin mambalgin-2, antagonist of ASIC1a, inhibited nocistatin-induced current, therefore the close similarity of mechanisms for ASIC1a activation by peptide and protons could be suggested. Thus, nocistatin is the first endogenous direct agonist of ASICs. This data could give a key to understanding ASICs activation regulation in the nervous system and also could be used to develop new drugs to treat pathological processes associated with ASICs activation, such as neurodegeneration, inflammation, and pain.

## 1. Introduction

Acid-sensing ion channels (ASICs) are members of ligand-gated cation channels and belong to the degenerin/epithelial Na^+^ channel superfamily. Four genes (ACCN1–ACCN4) express six subunits (ASIC1a, ASIC1b, ASIC2a, ASIC2b, ASIC3, and ASIC4), four of which can form functional homomeric channels (ASIC1a, ASIC1b, ASIC2a, and ASIC3) [1,2]. ASIC channels are widely expressed throughout the neurons of central and peripheral nervous systems, where they play a role in neurotransmission, synaptic plasticity, and learning, and in ischemia, neuronal cell death. ASICs are responsible for the sensitivity to the spontaneous, post-operative, neuropathic, inflammatory pain perception, and allodynia in a model of migraine [3,4,5,6,7,8,9,10]. ASIC isoforms have different proton sensitivity; so, a pH for half-maximal activation (pH_50_) for ASIC1a is 6.4–6.6, for ASIC1b, 5.9–6.3, for ASIC2a, 4.3–4.9, and for ASIC3, 6.5–6.7 [11,12,13,14,15,16].

In addition to protons, several other ASICs activators are known: The coral snake toxin MitTx, acting at nanomolar concentrations on ASIC1a and ASIC1b and at micromolar concentrations on ASIC3 [17]; a synthetic compound, 2-guanidine-4-methylquinazoline (GMQ), acting on ASIC3 at pH < 7.4 through a nonproton sensor [18] and also modulating delayed rectifying K^+^, voltage-gated Na^+^, and l-type Ca^2+^ currents [19]; lindoldhamine, acting as an agonist and positive allosteric modulator of human and rat ASIC3 channels in a broad range of pH, including pH > 7.4 [20]; endogenous lysophosphatidylcholine (LPC) or arachidonic acid, evoking a constitutive depolarizing ASIC3-mediated current [21] and endogenous opioids precursors, tetrahydropapaveroline (THP) and reticuline, activating human and rat ASIC3 channels at physiological pH and preventing a steady-state desensitization of the channels [22].

The nocistatin in mammalian is expressed as a precursor protein coding three neuropeptides (nocistatin, nociceptin, and Orphanin FQ2). Two neuropeptides, nociceptin and Orphanin FQ2, do not have activating effects on ASIC channels (Appendix A and [23]). Nocistatin is well represented in different parts of the brain and spinal cord [24]. However, it also was found in the peripheral nervous system (PNS), where the expression of its precursor significantly increases during the inflammation [25]. Nocistatin was studied in animal models as a prospective molecule together with opioid neuropeptides (see for review [26]). Nocistatin was shown to play an important role in the regulation of several physiological processes in which ASICs are also involved such as nociceptive transmission, learning, and memory processes in the central nervous system. In particular, nocistatin prevented the disruption of learning and memory processes in a scopolamine-induced impairment of learning and memory model [27] and produced anxiogenic-like actions in the elevated plus-maze test in mice [28]. In PNS, nocistatin may play a dual role. On one hand, nocistatin doses of 0.5–500 pg per mouse eliminated allodynia and thermal hyperalgesia effects caused by the action of nociceptin [29]. On the other hand, i.pl. injection of nocistatin in doses of 0.01–1 pmol per mouse induced pronociceptive responses in an algogenic-induced nociceptive flexion test in mice [30]. Up to date at least two nocistatin binding proteins have been reported on neurons: Unidentified G-coupled receptor(s) opening of canonical transient receptor potential cation (TRPC) channels through G_αq/11_-phospholipase C-protein kinase C pathway [31], and NIPSNAP1, a protein of unknown function, although it is predominantly expressed in the brain, spinal cord, liver, and kidney [32].

In this study, we identified rat nocistatin (rNS) as the first endogenous neuropeptide, which activates all the functional isoforms of rat ASICs channels.

## 2. Materials and Methods

### 2.1. Chemical Synthesis of Nocistatin

Nocistatin from rats (35 amino acids in length) having the sequence MPRVRSVVQARDAEPEADAEPVADEADEVEQKQLQ (Mw = 3907 Da) was synthetized by a solid-phase peptide synthesis technique on an automatic peptide synthesizer from Fmoc-protected amino acids (Iris Biotech GmbH, Germany) using 2-(7-Aza-1*H*-benzotriazole-1-yl)-1,1,3,3-tetramethyluronium hexafluorophosphate as a condensing agent. A C-terminal amino acid was attached to the Tentagel HL-NH 2 resin and modified with carboxy trytil linker (Tentagel-TRT, Germany) in the presence of *n,n*-Diisopropylethylamine. An eight-fold excess of amino acid per step was used, and in total, 3 mg of crude final product was obtained after cleavage and deprotection by Trifluoroacetic acid /DL-Dithiothreitol /H_2_O/ Di-n-butylmagnesium 150/4/3/0.5 (by weight) mixture. About 2 mg of pure nocistatin were obtained after RP-HPLC on a Triart-C18 10u 30 by 150 mm column (YMC, Kyoto, Japan) in an acetonitrile/water gradient, and the quality of the peptide was checked by ESI-MS. All reagents and solvents used without additional purification were purchased from Acros Organics (Thermo Fisher Scientific, New Jersey, NJ, USA) and Sigma-Aldrich (Merck KGaA, Darmstadt, Germany).

### 2.2. Electrophysiological Studies on Xenopus Laevis Oocytes

Oocytes expressed rASIC1a, rASIC1b, rASIC2a, and rASIC3 homomeric channels were prepared as described [20]. Injected oocytes were kept for two–three days at 19 °C and then for up to seven days at 15 °C in ND-96 medium supplemented with gentamycin (50 μg·mL^−1^) and containing (in mM): 96 NaCl, 2 KCl, 1.8 CaCl_2_, 1 MgCl_2_, and 10 HEPES titrated to pH 7.4 with NaOH. Electrophysiological recordings were performed using the GeneClamp500 amplifier (Axon Instruments, Inverurie, UK) at a holding potential of -50 mV. The data were filtered at 20 Hz and digitized at 100 Hz using the L780 AD converter (L-Card, Moscow, Russia). Microelectrodes were filled with 3 M KCl. The external bath solution was ND-96 with pH adjusted to 7.4/7.8 (10 mM HEPES) /8.5 (10 mM Tris). Nocistatin containing solutions were freshly prepared and buffered each experimental day in ND-96 bath solution (pH 7.4/7.8/8.5). Proton-activated currents through ASIC channels were elicited by application of ND-96, in which 10 mM of HEPES was substituted with 10 mM MES at pH 5.5/ 10 mM acetic acid at pH 4.5. The whole-cell recording was performed if a response to the control pH stimulus was stable in at least two replicates. Nocistatin-activated currents were elicited by application of nocistatin containing solutions with various concentrations of the peptide.

Extracellular solutions were applied using a computer-controlled valve system with a flow rate of 1 mL/min. The oocyte chamber volume was 80 µL. The validity of the application system for studying ASIC channels expressed in oocytes was confirmed in experiments by application of a Cl-free solution (containing (in mM): 130 NMDG, 1 MgGluc2, 2 KGluc, 5 HEPES) (pH 7.4). Fast application of proton/peptide-activating solution was performed at a flow rate of 1 mL/sec (the change of the bath solution occurred within 100 ms), followed by PC controlled flow stopping (from 5 to 15 s).

### 2.3. Data and Statistical Analysis

Analysis of electrophysiological data was performed using OriginPro 8.6 software (OriginLab, Northampton, MA, USA). Curves were fitted using different logistic equations:

(**a**) Usual logistic equation [33] F_1_(*x*) = A – A/(1 + (*x/x*_0_)^nH^), where F_1_(x) is the current amplitude at nocistatin concentration *x* ([*x*]) (response at concentration x), A is the maximal current amplitude (or the maximum asymptote or the stabilized response for an infinite concentration), *x*_0_ is the value of [*x*] at which the function reach half their maximum value (the EC_50_ value), and nH is the Hill coefficient (slope factor or a level of cooperativity);

(**b**) Equation for the model of unequal sites occupation [34] F_2_(*x*) = A/(1 + (x/EC_50_1)^nH^)*(1 + (*x*/EC_50_2)), multiplication of two logistic equations where F_2_(*x*) is the current amplitude at nocistatin concentration x, A is the maximal current amplitude, [EC_50_1] is the half-maximal concentration of nocistatin binding to pool of “cooperative” sites (or 0.5 probability of “cooperative” sites’ occupancy) and [EC_50_2] is the half-maximal concentration of nocistatin binding to additional unequal site (or 0.5 probability of single site occupancy), and nH is the Hill coefficient (slope factor) for the pool of cooperative sites;

(**c**) Generalized logistic differential equation (the four-parameter logistic nonlinear regression model or a Hill function) [35] F_3_(*x*) = ((A1 – A2)/(1 + (*x*/*x*_0_)^nH^)) + A2, where F_3_(x) is the response value at a given nocistatin concentration, x is the concentration of nocistatin; A1 is the control response value (fixed at 100%) or the minimum asymptote of the response when concentration = 0; A2 is the response value at maximal inhibition or the maximum asymptote or the stabilized response for an infinite concentration (% of control), x0 is the concentration at which 50% of the channels are expected to show the response (or the response halfway between the minimum response asymptote A1 and the maximum response asymptote A2) (IC_50_ value) and n_H_ is the Hill coefficient (a level of cooperativity). The maximum amplitude (I_max_) was calculated for each oocyte by individual fitting; the data were normalized to it, then the normalized data were averaged and fitted by the logistic equations F_1_(*x*) and F_2_(*x*).

(**d**) The time course of curves’ decay was fitted using a mono-exponential equation: F_4_(*x*) = A1*e^(-x/τdes)^ + A0, where F_4_(*x*) is the current amplitude at the time point x, A1 is the maximal current amplitude, A0 is its steady-state value and τ_des_ is the rate of desensitization.

All data are presented as mean ± SEM. The significance of normalized data differences was determined with a non-parametric Kruskal-Wallis ANOVA test with the significance level * *p* < 0.05 and ** *p* < 0.01; ns, non-significant.

### 2.4. Ethics Statement

This study was carried out in strict accordance with the World Health Organization’s International Guiding Principles for Biomedical Research Involving Animals. The protocol was approved by the Institutional Policy on the Use of Laboratory Animals of the Shemyakin-Ovchinnikov Institute of Bioorganic Chemistry RAS (Protocol Number: 251/2018 26.02.18).

## 3. Results

### 3.1. Nocistatin Directly Activates ASICs the Same As Protons

The ability of nocistatin to activate ASIC channels was studied in whole-cell configuration on oocytes of *Xenopus laevis*. We found an intriguing fact on cells expressing different ASICs isoforms, that nocistatin application at resting pH 7.4 causes rapidly activating inward currents in homomeric ASICs (Figure 1a–d). The shape of these currents was similar to currents caused by the rapid decrease of pH in the extracellular solution. Nocistatin induced currents that were fast activated and fast inactivated and included transient and sustained components of current in the case of ASIC2 and ASIC3—the same as the proton-activated one. The channels did not respond further to rapid acidification after activation by the peptide. Nocistatin likely shares mechanisms of activation and desensitization with protons, as can result from binding sites overlapping. However, the response of the channels to a pH drop was completely restored after washing out the peptide. In control experiments, nocistatin did not produce any effects on the uninjected oocytes (Figure 1e).

We compared kinetic parameters of ASICs responses to the protons versus nocistatin activation (Table 1). The currents were analyzed by three parameters: The peak amplitude value (A_current_), the time necessary to reach the maximum amplitude (t_max_), and the rate of desensitization (τ_des_ or the desensitization time constant). The A_current_ for ASIC1a and ASIC1b did not differ between protons and nocistatin. ASIC2a’s maximal amplitude of peptide activation was eight times higher than the response to a standard pH drop—7.4 to 5.5—therefore, nocistatin is a more effective ASIC2a activator compared to moderate acidification (and even in comparison with a stronger acidic stimulus, such as a pH drop 7.4 to 4.5 (Table 1)).

ASIC3′s response to nocistatin was slightly less (at 18%) than the response to protons, but this difference was statistically significant (Figure 1f). The t_max_ parameter showed no differences for ASIC1a and ASIC1b currents (Figure 1g). The response of ASIC2a to nocistatin was faster than that to acidification (t_max_ ~37% less); i.e., peptides produce channel openings more effectively. ASIC3 activation by the nocistatin was slower (t_max_ ~ 49% greater) (Figure 1g). We found that nocistatin-induced currents of ASIC1a desensitized significantly more quickly (τ_des_ difference 41%). ASIC3 currents induced by nocistatin, on the contrary, desensitized more slowly (τ_des_ difference 32%). No differences were observed in desensitization time between stimuli for ASIC1b (Figure 1h).

Overall, nocistatin at 1 mM can induce currents through ASICs that are identical to proton-induced currents in the case of the ASIC1b subtype, but current parameters have some minor variations for other ASIC isoforms (Figure 1a–d). Nocistatin caused a more quickly desensitizing current on ASIC1a. On ASIC2a, nocistatin causes a more rapidly activated current with a larger amplitude than the response to a pH drop of 7.45.5. Nocistatin caused a more slowly activated and desensitizing current with a tiny decrease of amplitude on ASIC3.

### 3.2. Effect of Nocistatin on Rat ASIC1a Channels

For a more detailed study of nocistatin action on ASICs, we used the ASIC1a isoform as one of the most important and abundantly represented in the central nervous system [36,37,38]. At extracellular pH 7.4, nocistatin dose-dependently activated ASIC1a (Figure 2a). Subsequent pH stimulus manifested the rest of unaffected ASIC1a channels; this number was also dependent on the concentration of the peptide applied (Figure 2a). The maximal activating effect by 1 mM nocistatin was comparable to the response to a pH drop of 7.4–5.5 in the control experiment. Moreover 1 mM nocistatin retained the ability to effectively activate the ASIC channel even at more alkaline conditioning pH, such as 7.8 and 8.5 (Figure 2b). ASIC1a current amplitude was 95.5 ± 1.8, 89.5 ± 2.8 and 86.7 ± 2.5% to one measured for the control pH 5.5 stimulus from respectively conditioning pH 7.4, 7.8 and 8.5. To quantitatively describe the activating effect of the peptide, we used the usual logistic equation F_1_, which is used to describe the activating/potentiating effect of ligands on receptors, including ASIC channels (see Materials and Methods; and [39]). The fitting of the peptide activation curve by the usual logistic equation (F_1_) produced an unacceptable result (Figure 2c, dashed line), so the channel’s activating dependence was analyzed by a modified equation (F_2_) (see Material and Methods for detailed description) (Figure 2c, solid line; Appendix A). Equations (F_2_) can take into consideration the presence in the channel of several binding sites with different affinities: The pool of cooperative sites with the half-maximal effective concentration 1 (EC_50_1) and a constant and additional independent single site with the half-maximal effective concentration 2 (EC_50_2) constant (see Materials and Methods; and [34]). The data were well-fitted to the equation (F_2_), calculated EC_50_1 and EC_50_2 values were 0.247 ± 0.025 and 0.14 ± 0.07 mM, respectively, and the Hill coefficient (n_H_) was 7.44 ± 0.95 (*n* = 5).

The response current of ASIC1a to protons’ application after 0.125–0.5 mM nocistatin pretreatment was split in two peak components (Figure 2a). These proton-induced currents showed peak amplitude maximums at t_max_1 = 495 ± 85 ms for 1st and t_max_2 = 1415 ± 112 ms for 2nd components of the rASIC1a current (Figure 2a). To quantitatively describe the inhibitory effect of the peptide, we used the Hill function, which is usually used to describe the inhibitory effect of ligands on receptors, including ASIC channels (see Materials and Methods; and [35,40]). Fitting of the amplitude’s decreasing effect of nocistatin pretreatment for the 1st and 2nd components of the current with the logistic equation (F_3_) yielded IC_50_ values of 0.115 ± 0.016 mM (n_H_ = 3.5 ± 0.4) and 0.24 ± 0.02 mM (n_H_ = 1.9 ± 0.2) (*n* = 5), respectively (Figure 2d, blue and green lines; Appendix A). Thus, nocistatin reduced the 1st component of the current twice as effectively, and in the presence of 0.5 mM of the peptide, we can observe only the 2nd component.

Acid-induced steady-state desensitization (SSD) is a distinguishing characteristic of ASICs when low acidification doesn’t activate the channels but decrease further respond to the stimulus in a concentration-dependent manner. We examined whether nocistatin can induce SSD the same as protons (Figure 2e). When low concentrations of rNS (0.04 to 0.1 mM) were applied the response to the 1 mM nocistatin stimulus was the same as without rNS pre-application. Therefore, rNS does not -induce SSD.

To verify specific effects of nocistatin on rASIC1a, the inhibition of ASIC1a currents was studied using a toxin, mambalgin-2 (Mamb-2), a well-known effective inhibitor of ASIC1 channels [41]. At a concentration of 1 μM, Mamb-2 strongly inhibited the nocistatin-induced current (84.3 ± 0.7%) (*n* = 4) (Figure 2f). These data are identical to Mamb-2′s inhibiting effect on proton activation of ASIC1a expressed in *X. laevis* oocytes (83 ± 2.7%).

## 4. Discussion

All previously known activators of ASIC channels, both exogenous and endogenous, cause a persistent, slowly desensitized current that was different in kinetic parameters from currents induced by pH drops. For example, GMQ elicited slowly developing, non-desensitizing and non-selective cation current [14,18,19]. We found that protons and nocistatin can operate ASICs in a similar manner. Both agonists show rapid kinetics of desensitization of the currents and have a high Hill coefficient of activation. The identical efficiency with which the toxin Mamb-2 inhibits the nocistatin- and proton-induced activation of ASIC1a (Figure 2f), as well as the similarity of the curves of nocistatin- and proton-dependent activation of the channel [34], indicate a similarity of the mechanisms of these activations, although there are quite significant differences in desensitization rates. The striking identity of the nocistatin- and proton-induced currents in the case of ASIC1b allows us to conclude that the functioning (activation and inactivation) of this isoform is completely identical both in the presence of nocistatin and protons. The most significant differences were observed in the case of ASIC2a, however, this isoform differs greatly from other functional isoforms by its low sensitivity to pH change.

As was shown on chicken ASIC1 channel, the decrease of extracellular pH results in the collapse of the acidic pocket followed by reorganization of the extracellular domains, which leads to rotation of the subunits, a shift of the lower palm domain towards the membrane and the channel opening. The subsequent process of desensitization involves the further rearrangement of the lower palm domain, which leads to the restitution of transmembrane domains to a resting-like conformation and while β11-β12 linkers decouple the collapsed acidic pocket from the channel pore resulting in the non-conducting ion channel in the presence of the activator [42]. Thus, we can assume that the activation of the channel by nocistatin can affect the acidic pocket or any other domains involved in signal transduction from acidic pocket to pore region that leads to the inability of the channels to be activated by protons (Figure 1). Relatively large size of the peptide may be the cause of a significantly greater influence on the reorganization of domains and therefore cause faster and more effective desensitization of the ASIC channels in comparison to the protons.

Expression of nocistatin precursor is rather high in CNS therefore this peptide can take part in regulation of ASICs activity in brain. Some mathematical calculation indicates that nocistatin can affect endogenous ASIC channels. ASICs in CNS are mostly localized on postsynaptic membrane therefore high nocistatin concentrations for their activation should be achieved in synaptic cleft. According to the literature, the content of nocistatin in the mouse spinal cord was estimated at 0.06 pmol/mg, and its content may increase several times in pathological conditions [26,43]. About 10^−21^ mole nocistatin per synapse is required for the activation of ASICs if we consider the presynaptic vesicle volume to be approximately 5 × 10^5^ nm^3^ and take into account tenfold dilution in the synaptic cleft [44]. Since the total number of synapses in the brain was estimated at 10^15^ [45] as well as nocistatin obviously cannot be expressed in every neuron, the concentration of nocistatin sufficient for ASICs activation can be achievable. Rapid elevation of expression of nocistatin precursor in PNS also takes place during inflammation [25], highlighting the role of high concentration of this peptide in initiation of inflammation. Compartmentalization of neurons in CNS and PNS enables the production of high concentrations of neuromodulators necessary for ASICs regulation in vivo. We showed that nocistatin has a complex modulating effect on ASIC channels: On the one hand, it acts as an inhibitor in low concentrations (0.2 mM or less) (Figure 2d; Appendix A), and on the other hand, nocistatin can independently activate the channels at high concentrations. This experimental fact may partly explain the bidirectional role shown for nocistatin in in vivo experiments (i.e., participation in nociceptive and anti-nociceptive processes), although the significance of ASICs regulation by nocistatin in neurons of the central and peripheral nervous systems needs to be resolved in further studies. However, it is worth noting that the presence of other potential molecular targets for nocistatin was reported [31,32]. This is usual for endogenous neuropeptides to affect several receptors/channels to produce a distinct response to various concentrations of ligands. For example, the calcitonin gene-related peptide, dynorphins, and enkephalins interact with several distinct receptors [9,46,47]. Neuropeptides like opioid peptides (dynorphins, enkephalins) and nocistatin influence on processes of algesia and analgesia and should be considered as bidirectional regulatory molecules.

## 5. Conclusions

Nocistatin, an endogenous neuropeptide, is the first direct ASIC channels agonist. The nocistatin- and proton-induced currents almost completely coincide, which may be evidence of similar mechanisms controlling the functioning of the channels. At the same time, nocistatin has a bi-directional in vivo action. Therefore, nocistatin and factors that can regulate its production or elimination could be included in consideration by investigations of normal and pathological processes involving ASICs. Moreover, regulation of nocistatin production could be a good pharmacological strategy for treatment of inflammatory and neurodegenerative diseases.

## Figures and Tables

**Figure 1 biomolecules-09-00401-f001:**
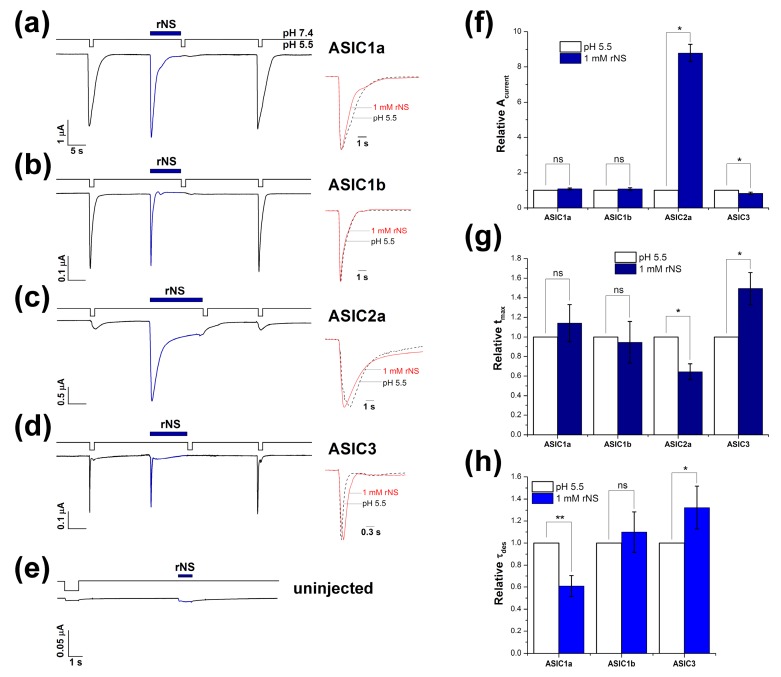
Activation of rat acid-sensing ion channels by rat endogenous neuropeptide nocistatin (rNS). Actions of 1 mM rNS on acid-sensing ion channels (ASIC1)a (**a**), ASIC1b (**b**), ASIC2a, (**c**) and ASIC3 (**d**) expressed in oocytes are shown together with responses to control stimuli. (**e**) illustrates application of rNS and pH 5.5 stimuli on uninjected oocytes (negative control). Currents were measured at a conditioning pH 7.4 in an external bath and were compared to control currents induced by a pH drop to 5.5 on the same cell. The kinetics variation of ASICs response to the peptide (red line) and pH 5.5 (dashed line) stimulus: (**f**) variation of the peak amplitude value (A_current_), (**g**) variation in time to reach the maximal amplitude (t_max_) and (**h**) variation of the desensitization time constant (τ_des_). Each bar is presented as mean ± SE of 4–7 measurements. * *P* < 0.05 and ** *P* < 0.01 are significantly different values than the one measured for control current at pH 7.4-5.5 drop; ns, non-significant difference; Kruskal–Wallis ANOVA.

**Figure 2 biomolecules-09-00401-f002:**
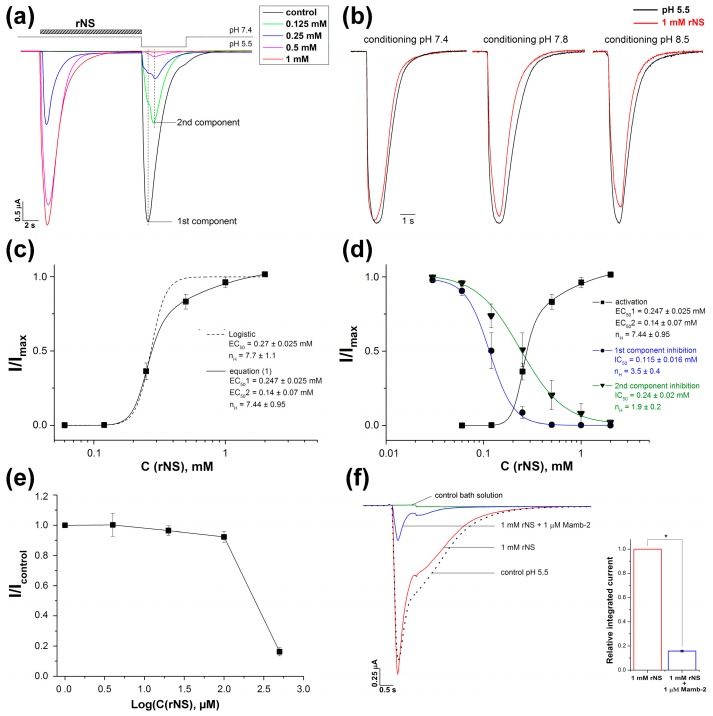
Effects of nocistatin on rat ASIC1a channels. (**a**) Whole-cell currents were obtained as a response to various concentrations of rNS, followed by a pH-5.5 stimulus that activated the rest of the ASIC1a channels unaffected by rNS. Dotted lines mark the maximums of the 1st and 2nd components of the pH 5.5-activated current (*n* = 5). (**b**) Normalized representative traces of ASIC1a currents evoked by pH 5.5 stimulus (black) and 1 mM rNS (red) at various conditioning pH indicating that rNS retains the ability to the effective activation of ASIC channel even in an alkaline medium (*n* = 4). (**c**) Dose dependence of ASIC1a activation by rNS fitted by a usual logistic equation (dashed line) and by equation F_2_(x) (solid line). Calculated parameters of EC_50_ for non-discriminated sites, or EC_50_1 for “cooperative” sites and EC_50_2 for single site, are shown as a result of experimental data mathematical processing. The peak current amplitude is normalized to the maximum amplitude (I_max_) calculated for each oocyte by individual fitting. Data are presented as mean ± SEM; *n* = 5. (**d**) Dose-dependent amplitude decreases in currents through ASIC1a in response to pH 5.5 stimulus following rNS action. Two independent dose-response curves were measured for the 1st (blue line) and the 2nd (green line) peak components of the current. Calculated parameters of IC_50_ are shown as a result of experimental data mathematical processing by F_3_(x) logistic equation (see Materials and Methods for detailed description). Peak current amplitudes are normalized to the maximum amplitude (I_max_) of pH 5.5-induced current. The dose response curve of ASIC1a activation by rNS (black line) is equal to that presented in panel B. Data are presented as mean ± SEM; *n* = 5. (**e**) Dose-response curve of ASIC1a steady-state desensitization (SSD) initiated by rNS preconditioning. The peak current amplitude of ASIC1a incubated with various conditioning rNS concentrations was measured at 1 mM rNS stimulus, and was normalized to the amplitude of the control current evoked by 1 mM rNS without the preincubation. rNS showed no SSD effect up to concentrations of the peptide, activating the channels. Data are mean ± SEM (*n* = 4). (**f**) Representative traces of ASIC1a currents evoked by pH 5.5 stimulus (dotted line), 1 mM rNS (red trace) and 1 mM rNS in the presence of 1 μM Mambalgin-2 (Mamb-2) (blue trace); control bath solution (green trace) and an integral current plot for 1 mM rNS (control value) and 1 mM rNS mixed with 1 μM Mamb-2. All presented traces were obtained from the same cell. Each point is presented as mean ± SE of 4 measurements. * *p* < 0.05, Kruskal–Wallis ANOVA.

**Table 1 biomolecules-09-00401-t001:** Experimental data measured for homomeric rat ASIC channels expressed in oocytes.

Isoform	Amplitude, nA		t_max_, s		τ_des_, s	
	pH 5.5	rNS	pH 5.5	rNS	pH 5.5	rNS
**ASIC1a**	3300 ± 522	3525 ± 497	0.47 ± 0.08	0.52 ± 0.11	2.6 ± 0.4	1.4 ± 0.1
**ASIC1b**	762 ± 212	869 ± 285	0.32 ± 0.06	0.27 ± 0.03	0.51 ± 0.09	0.5 ± 0.06
**ASIC2a**	182 ± 20 (1039 ± 43) ^1^	1609 ± 246	1.45 ± 0.19 (1.27 ± 0.15) ^1^	0.94 ± 0.19	ND (7.6 ± 0.5) ^1^	2.22 ± 0.16
**ASIC3**	210 ± 62	165 ± 46	0.17 ± 0.02	0.25 ± 0.03	0.33 ± 0.09	0.4 ± 0.1

The peaks’ amplitude—t_max_ as the time to reach the amplitudes maximum and τ_des_ as the desensitization time constant—were determined in whole-cell configuration for channels activated by fast acidification (pH drop 7.4–5.5) and by 1 mM rNS application. ND: Not determined. Data are presented as mean ± S.E. from 4–7 independent measurements. ^1^ data obtained for pH drop 7.4–4.5.

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
