# Peer review of "Endogenous Neuropeptide Nocistatin Is a Direct Agonist of Acid-Sensing Ion Channels (ASIC1, ASIC2 and ASIC3)"

_biomolecules, 2019, doi:10.3390/biom9090401_

Round 1
Reviewer 1 Report
In this manuscript, the authors present data supporting the discovery of the endogenous peptide nocistatin as a novel agonist of ASIC channels, the first one that would activate the channels in a similar manner than protons (transient current).
Despite the originality and the novelty that would constitute such a discovery in the physiopathological involvements of ASIC channels in central as well as peripheral nervous system, the impact of this manuscript is weakened by the quality and the quantity (n values are small) of the data, that do not fully support the ambitious conclusion drawn. A major revision is needed.
Introduction, line 36- In the pain field, and linked to their expression in sensory neurons, ASIC channels were not only described to be involved in inflammatory pain, but also in acute, neuropathic, joint or migraine pain, for exemple.
Line 50- The paper cited as ref 20 only tested nociceptin by not orphanin. It could be interesting to shown your personal negative data to support the specificity of nocistatin on ASICs, compared to the other two peptides derived from the same precursor, but also in the same time, to provide a negative control to validate your experimental protocol on heterologous currents expressed in Xenopus oocytes with other peptides than NS (see following question..).
Line 94- The data shown in both Fig1 and 2 would be mainly the same if ASIC channels were activated by protons and not by NS. Thus, accurate controls of the pH value of solutions containing NS are required and this should be extensively explained (and shown) to eliminate the possibility of proton-induced activation. What was the vehicle of stock NS solutions? Were the pH checked after the dilution of NS in ND96? Were they also checked after the electrophysiological experiment (acidification during the experiment?)?...
Figure 1- What seems to happen is that 1 mM NS activate ASICs like would do a maximal activation by protons. But it was only compared with pH 5.5-induced current, that could be the maximal current for ASIC1a, ASIC1b and ASIC3, but not for ASIC2a channels. The comparison with maximal proton-activated ASIC2a current is needed and not only with pH 5.5 and the conclusions need to be adapted. The data presented in the Table 1 (with values at pH 4.5 for ASIC2a) also go in the same direction…
Line 178- Linked to the previous remark, the sentence “The response of ASIC2a to nocistatin was faster than that to acidification (tmax ~37% less); i.e., peptides produce channel openings more effectively” is probably abusive. A comparison with the maximal proton-gated ASIC2a current is needed to conclude.
Line 190- Based on the previous result described in line 178 about putative effect of NS on ASIC2a current, it feels strange to focus on ASIC1a current. This is justified by saying “we used the ASIC1a isoform as one of the most important and abundantly represented in the central nervous system”, but the most abundantly channels in the CNS are not homomeric ASIC1a channels, but heteromeric ASIC1a+ASIC2-containing channels.
What are the effects of NS on heteromeric channels that have been shown to be the most relevant forms in central as well as peripheral neurons? Additionnal data on NS effects on native ASIC currents recorded from central or sensory neurons in culture would also strengthen the relevance of this new regulation.
Line 193 and Figure 2A- “Subsequent pH stimulus manifested the rest of the ASIC1a channels unaffected by NS”: this is not shown by your data. In figure 2A, there is apparently no additivity of the amplitude the NS-activated current and the subsequent proton-activated current to reach the same amplitude of the control maximal NS-activated or proton-activated current, when looking at the green and blue traces, like would have been expected. About the green trace, is that normal that 0.125 mM NS activates no ASIC current?
By the way, to test the additivity of the two currents and support a same pool of channels activated by NS or protons, it would have been more accurate to compared the total current (quantity of charges flowing through the channels) calculated as an area under curve. And particularly because of the “problem” of double peak that make difficult to measure a maximal amplitude…). What was the number of experiments for this protocol (Fig. 2A)?
The reverse experiment needs also to be done: first activate ASIC current with protons (non maximal activation, for ex. at pH50) and show that the subsequent NS-activated current is reduced proportionally.
Line 196 and Figure 2B- Why only testing NS effects at alkaline pH? What would be the effect of acidic conditioning pH on NS-activated current (see also the experiment suggested above). What are the effects of nocistatin on pH-dependent activation and inactivation curves of ASIC? Several known activator of ASIC channels have already been shown to induce some shifts of these curves, thus inducing channel openings at pH 7.4.
Figure 2C and D- For the dose response curve, as well as the subsequent effect on proton-activated current, the analysis should be redone with the total current (area under cuve).
Figure 2E and line 246- The authors conclude that there is no desensitization induced by NS, but the transient kinetic of the current activated by NS suggests the opposite. Indeed, the current inactivates during a sustained application of NS. The experiment showing that there would be a same pool of channels activated by NS and protons also support a desensitization of the channels by NS like by protons. This needs to be clarified.
Figure 2F- What is the purpose of testing Mambalgin? The data only show that the current activated by NS is an ASIC(1a) current. But what else could it be in Xenopus oocytes injected with only ASIC1a? These data bring nothing about the “molecular” way of action of NS compared to mambalgin.
Discussion – Based on the data, it is impossible to postulate on the molecular way of action of NS compared to protons or mambalgin. The discussion needs to be refocused on the results, unless providing new data with a molecular approach (structure-function data etc…).
Line 284- Based on the results, it is difficult to postulate on their physiopathological relevance in pain sensing: activation of ASIC channels is expected to induce pain, but inactivation by sustained presence of nocistatin would impaired a subsequent activation by protons, thus reducing pain. This need to be discussed together with the “bidirectional” in vivo effects described by the bibliography (and detailed in the introduction lines 59-63). Supplementary in vivo data could also be provided testing NS effects on ASIC-dependent pain models and/or on ASIC KO mice, to support a physiological relevance of NS effect on ASIC channels.
Conclusion line 288- There seems to be a problem in this sentence: “Currents activated by this peptide, convergence with those activated by protons.”
Author Response
Point 1: Introduction, line 36- In the pain field, and linked to their expression in sensory neurons, ASIC channels were not only described to be involved in inflammatory pain but also in acute, neuropathic, joint or migraine pain, for example.
Response 1: Additional information about participation in pain sensitivity was added (lines 36-37). Thank you for this recommendation.
Point 2: Line 50- The paper cited as ref 20 only tested nociceptin by not orphanin. It could be interesting to show your personal negative data to support the specificity of nocistatin on ASICs, compared to the other two peptides derived from the same precursor, but also in the same time, to provide a negative control to validate your experimental protocol on heterologous currents expressed in Xenopus oocytes with other peptides than NS (see following question..).
Response 2: We added the controls with nociceptin and orphanin (please, see Supplementary data, Suppl. Figure 1). We would also like to note that the paper Ref.20 tested orphanin under a Neuropeptide2 name (NocII) (Table 1, Mix 9).
Point 3: Line 94- The data shown in both Fig1 and 2 would be mainly the same if ASIC channels were activated by protons and not by NS. Thus, accurate controls of the pH value of solutions containing NS are required and this should be extensively explained (and shown) to eliminate the possibility of proton-induced activation. What was the vehicle of stock NS solutions? Were the pH checked after the dilution of NS in ND96? Were they also checked after the electrophysiological experiment (acidification during the experiment?)?...
Response 3: All solutions were prepared immediately before the measurements. Nocistatin-containing solutions and control solutions (ND96) only differed in the presence or absence of the peptide. The purity of the peptide was approximately 98% (we indicated this in the Materials and methods). The maximal concentration of the nocistatin-containing solution was 2 mM wherein no change of pH was detected. Additionally, the pure effect of nocistatin was confirmed in the experiment of ASIC1a activation using more alkaline conditioning pH. Nocistatin retained the ability to activate the ASIC channel effectively even at conditioning pH 8.5, which suggest an absence of proton-induced activation via a shift of pH sensitivity of the channel. Application of the bath solution instead of the acid stimulus (green line, Figure 2F) served as a control of the absence of acidification during the experiment.
Point 4: Figure 1- What seems to happen is that 1 mM NS activate ASICs like would do a maximal activation by protons. But it was only compared with pH 5.5-induced current, that could be the maximal current for ASIC1a, ASIC1b and ASIC3, but not for ASIC2a channels. The comparison with maximal proton-activated ASIC2a current is needed and not only with pH 5.5 and the conclusions need to be adapted. The data presented in the Table 1 (with values at pH 4.5 for ASIC2a) also go in the same direction…
Response 4: All channels' activations were compared with pH 5.5 stimulus for uniformity and in consideration of the serious physiological limitation for stimuli below pH 5.5 in vivo (Gründer and Pusch, 2015). At the same time, according to the literature, the content of nocistatin in the mouse spinal cord is estimated at 0.06 pmol/mg, and its content may increase several times in pathological conditions (Okuda-Ashitaka and Ito, 2015; Liu et al., Neurosci lett 2012). If we consider the presynaptic vesicle volume to be approximate 5x10^5 nm^3 and take into account tenfold dilution in the synaptic cleft (Gründer & Pusch, 2015), then it can be assumed that the order of 10^(-23)-10^(-24) mol nocistatin per synapse is required for the activation of ASICs, which is approximately 10^11 times less than the total estimated physiological content of nocistatin. The total number of synapses in the brain is estimated at 10^(15) (Nguyen, Undergraduate Journal of Mathematical Modeling 2010). Thus, the concentration of nocistatin, sufficient to activate ASICs, can be achievable, since it is obvious that nocistatin could be expressed in specific parts of the nervous system. For ASIC2a we compared the parameters of nocistatin-activated current with the current activated by pH 4.5 and showed that the kinetic parameters of these currents are still significantly different. A more detailed study of the action of nocistatin on ASIC2a is required including a comparison with the action of protons in a wider range of pH.
Point 5: Line 178- Linked to the previous remark, the sentence “The response of ASIC2a to nocistatin was faster than that to acidification (tmax ~37% less); i.e., peptides produce channel openings more effectively” is probably abusive. A comparison with the maximal proton-gated ASIC2a current is needed to conclude.
Response 5: Above in the text of the manuscript, we clarify that the term “more effectively" means the better than moderate acidification (pH 5.5) that is related to the physiological level. We do not deny that with strong acidification (pH 4 and lower), the parameters we have chosen for the nocistatin and proton-induced currents of ASIC2a channel converge, or even coincide.
Point 6: Line 190- Based on the previous result described in line 178 about putative effect of NS on ASIC2a current, it feels strange to focus on ASIC1a current. This is justified by saying “we used the ASIC1a isoform as one of the most important and abundantly represented in the central nervous system”, but the most abundantly channels in the CNS are not homomeric ASIC1a channels, but heteromeric ASIC1a+ASIC2-containing channels.
What are the effects of NS on heteromeric channels that have been shown to be the most relevant forms in central as well as peripheral neurons? Additionnal data on NS effects on native ASIC currents recorded from central or sensory neurons in culture would also strengthen the relevance of this new regulation.
Response 6: ASIC2a channel, due to its low pH sensitivity, stands apart and, in our opinion, deserves a separate study. ASIC1a channel is one of the most widespread and studied channels among ASICs. Nevertheless, we found for this channel rather non-trivial result (non-sigmoidal shape of the activation curve, separation of the pH-induced current into two components in the presence of nocistatin). In this Brief Report, we wanted for the first time to demonstrate the effect of the peptide specifically on homomeric channels. Undoubtedly, a more comprehensive study of the phenomenon of the action of nocistatin not only on homomeric, but also on heteromeric channels is required. In the latter case, the task is complicated by the fact that ASIC2b isoform, as well as the stoichiometry of heteromeric channels, must be taken into account.
Nocistatin has at least two established targets on the postsynaptic membrane - these are unidentified G-coupled receptor(s) opening canonical transient receptor potential cation (TRPC) channels through Gαq/11-phospholipase C-protein kinase C pathway (Chen et al., Neuroscience 2010), as well as 4-nitrophenylphosphatase domain and nonneuronal SNAP25-like protein homolog 1(NIPSNAP1), which binds nocistatin directly and inhibits TRPV6 activity (Okuda-Ashitaka et al., J Biol Chem 2012; Schoeber et al., Pflugers Arch. 2008) This fact does not change the significance of this manuscript findings, but, in our opinion, limits experimentation on neurons.
Point 7: Line 193 and Figure 2A- “Subsequent pH stimulus manifested the rest of the ASIC1a channels unaffected by NS”: this is not shown by your data. In figure 2A, there is apparently no additivity of the amplitude the NS-activated current and the subsequent proton-activated current to reach the same amplitude of the control maximal NS-activated or proton-activated current, when looking at the green and blue traces, like would have been expected. About the green trace, is that normal that 0.125 mM NS activates no ASIC current?
By the way, to test the additivity of the two currents and support a same pool of channels activated by NS or protons, it would have been more accurate to compared the total current (quantity of charges flowing through the channels) calculated as an area under curve. And particularly because of the “problem” of double peak that make difficult to measure a maximal amplitude…). What was the number of experiments for this protocol (Fig. 2A)?
The reverse experiment needs also to be done: first activate ASIC current with protons (non maximal activation, for ex. at pH50) and show that the subsequent NS-activated current is reduced proportionally.
Response 7: One of the distinguishing features of “fast channels” (it means rapidly activating and rapidly inactivating) like ASIC channels, is steady-state desensitization, i.e. the ability of channels to transfer into an inactive state in a low concentration of stimulus. Therefore there is a concentration of ligand that is not enough to activate the channels, but enough to inactivate them. When you decrease the conditioning pH from the resting pH 7.4, for example, to pH 7.1 and after a few seconds apply pH 5.5 the response would be significantly reduced. This is a standard and a common practice for measuring stationary desensitization (Waldmann, JBC 1997; Sherwood, JBC 2008; Alijevic, JBC 2012). In this case, since nocistatin acts by a mechanism similar to the action of protons, preapplication of the peptide at 0.125 mM concentration can also cause the steady-state desensitization of the channel without the activation.
We performed a recalculation for dose-dependences of activation and inhibition for the area under curve instead of current amplitude (Supplementary Figure 2 in the Supplementary Materials). No significant differences with the data shown in Figure 2C and 2D were found, except the inhibitory dose-dependence fitted worse for the total current. Figure S2B well-illustrates the answer to your question about the gap between summarised respond and control on Figure 2A. The black line on Figure S2B (first peak current (rNS)) and the blue line (second peak (protons)) curve summation gave the overall response curve for both stimuli.
The number of experiments for the protocol shown in Figure 2A was five (we indicated this in the caption to the figure).
In this Brief Research Report, we limited ourselves to data concerning only the activating action of nocistatin on ASICs as a most important effect first time described for a well-known endogenous peptide. However, more comprehensive studies of the interaction between the peptide and protons certainly need to be resolved in the future.
Point 8: Line 196 and Figure 2B- Why only testing NS effects at alkaline pH? What would be the effect of acidic conditioning pH on NS-activated current (see also the experiment suggested above). What are the effects of nocistatin on pH-dependent activation and inactivation curves of ASIC? Several known activator of ASIC channels have already been shown to induce some shifts of these curves, thus inducing channel openings at pH 7.4.
Response 8: Today it is known only GMQ that causes the opening of ASIC3 channel by the shift of activation and steady-state desensitization curves (Alijevic, JBC 2012). The other activator of ASIC3 lindoldhamine does not have such an effect (Osmakov, British Journal of Pharmacology 2018). For the peptide MitTx, which activates the ASICs (also demonstrated on oocytes), no information about its influence on pH-dependence was published (Bohlen et al., Nature 2011).
However, it is important to continue extensive and comprehensive studies of the effect of nocistatin on different functional homomeric and heteromeric channels and should include the pH-dependence experiments. In this Brief Report, we present only data regarding the channels direct activation by nocistatin.
Point 9: Figure 2C and D- For the dose response curve, as well as the subsequent effect on proton-activated current, the analysis should be redone with the total current (area under cuve).
Response 9: The recalculation for the total current was done and included in Supplementary Materials.
Point 10: Figure 2E and line 246- The authors conclude that there is no desensitization induced by NS, but the transient kinetic of the current activated by NS suggests the opposite. Indeed, the current inactivates during a sustained application of NS. The experiment showing that there would be a same pool of channels activated by NS and protons also support a desensitization of the channels by NS like by protons. This needs to be clarified.
Response 10: In this case, we described (lines 163-168 of the revised manuscript) steady-state desensitization and it was clearly stated in the text (we provided a detailed description in the text and above in response to point 7). We preincubated ASIC1a channel with nocistatin at concentrations, which are not yet able to cause activation of the channel (up to 0.125 mM), is capable to produce a response to further 1 mM nocistatin stimulus. The amplitude of 1 mM NS stimulus presented in Figure 2E, and this experiment not similar to one on Figure 2A, 2C, 2D (low concentrations of NS in preincubation together with pH5.5 stimulus).
Nocistatin, at concentrations up to 0.125 mM, did not cause significant inhibition of the subsequent nocistatin-induced current. In the case of proton-induced steady-state desensitization, the preincubation of ASIC1a channel with protons at a concentration of 0.00006 mM (or pH 7.2) leads to significant desensitization of the channel (Alijevic, JBC 2012; Besson, Neuropharmacology 2017). Thus we conclude that nocistatin does not cause steady-state desensitization of ASIC1a channel.
The legend to Figure 2E and corresponding text described this pretty well.
Point 11: Figure 2F- What is the purpose of testing Mambalgin? The data only show that the current activated by NS is an ASIC(1a) current. But what else could it be in Xenopus oocytes injected with the only ASIC1a? These data bring nothing about the “molecular” way of action of NS compared to mambalgin.
Response 11: The native Xenopus oocytes express various sodium channels (voltage-dependent, NH4Cl-sensitive, ATP-sensitive Na+-channels, etc.) as well as K+-channels, Cl--channels, Ca2+-channels, mechanosensitive and nonselective cation channels (Weber, J. Membrane Biol. 1999). However, the fact that mambalgin inhibited both nocistatin- and proton-induced currents with equal effectiveness allowed us to verify nocistatin activity at ASIC1a channel, and also to suggest the similar mechanisms of nocistatin-mediated and proton-mediated activation.
Point 12: Discussion – Based on the data, it is impossible to postulate on the molecular way of action of NS compared to protons or mambalgin. The discussion needs to be refocused on the results unless providing new data with a molecular approach (structure-function data etc…).
Line 284- Based on the results, it is difficult to postulate on their physiopathological relevance in pain-sensing: activation of ASIC channels is expected to induce pain, but inactivation by sustained presence of nocistatin would impair a subsequent activation by protons, thus reducing pain. This need to be discussed together with the “bidirectional” in vivo effects described by the bibliography (and detailed in the introduction lines 59-63). Supplementary in vivo data could also be provided testing NS effects on ASIC-dependent pain models and/or on ASIC KO mice, to support a physiological relevance of NS effect on ASIC channels.
Response 12: We have added in the Discussion section reasoning more focused on results (lines 279-287). Of course, nocistatin interaction with the channel at the molecular level is more likely speculative, but we believe that this hypothesis in the Discussion section may be interesting for readers. We also added a sentence about the possible correlation of the peptides’ bi-directional effect with in vivo experiments data (lines 305-311). We agree with you that adequate in vivo pain models, as well as experiments on knockout animals, would help understand the regulatory role of this endogenous peptide in the organism. However, effective and selective pharmacological blockage of additional rNs targets is not available (GPCR has not been identified yet) and it is almost impossible to propose an adequate in vivo model to verify effect only on ASICs.
Point 13: Conclusion line 288- There seems to be a problem in this sentence: “Currents activated by this peptide, convergence with those activated by protons.”
Response 13: We rephrased this sentence.
Reviewer 2 Report
The author show us the evidence of rat nocistatin as the first endogenous neuropeptide, which can directly activate different kinds of rat Asics chanels. The clarity of this work could be a good strategy for treatment of inflammatory and neurodegenerative diseases.
The only issue which I am considering is that is there any in vivo experiments to clarify the suspect?
Author Response
Point 1: The only issue which I am considering is that is there any in vivo experiments to clarify the suspect?
Response 1: To date in vivo studies are difficult to conduct due to the presence of earlier identified but poorly characterized molecular targets for nocistatin on neurons. Endogenous neuropeptides often have several molecular targets on neurons membrane to provide complex and controlled action. Nocistatin has at least two established targets on the postsynaptic membrane - these are unidentified G-coupled receptor(s) opening of canonical transient receptor potential cation (TRPC) channels through Gαq/11-phospholipase C-protein kinase C pathway (Chen et al., Neuroscience 2010), as well as 4-nitrophenylphosphatase domain and nonneuronal SNAP25-like protein homolog 1(NIPSNAP1), which binds nocistatin directly and inhibits TRPV6 activity (Okuda-Ashitaka et al., J. Biol. Chem. 2012; Schoeber et al., Pflugers Arch. 2008) This fact does not change the significance of this manuscript findings, but limits in vivo experiments by complexity of the system.
Reviewer 3 Report
see attached

Author Response
Point 1: The concentration of rat nocistatin (around 0.125-1 mM) used in this study tends to be very high. It remains to be determined whether such high concentrations would have physiological relevance in the central nervous system.
Response 1: Our data are limited to heterologous expression system and provide some calculations that nocistatin can affect endogenous ASICs channels. ASICs in CNS are mostly localized on postsynaptic membrane therefore high nocistatin concentrations for their activation should be achieved in the synaptic cleft. According to the published reports, the content of nocistatin in the mouse spinal cord is estimated at 0.06 pmol/mg, and its content may increase several times in pathological conditions (Okuda-Ashitaka and Ito, 2015; Liu et al., Neurosci lett 2012). If we consider the presynaptic vesicle volume to be approximate 5x10^5 nm^3 and take into account tenfold dilution in the synaptic cleft (Gründer & Pusch, 2015), then it can be assumed that the order of 10^(-23)-10^(-24) mol nocistatin per synapse is required for the activation of ASICs, which is approximately 10^11 times less than the total estimated physiological content of nocistatin. The total number of synapses in the brain is estimated at 10^(15) (Nguyen, Undergraduate Journal of Mathematical Modeling 2010). Thus, the concentration of nocistatin, sufficient to activate ASICs, can be achievable, since it is obvious that nocistatin is not expressed in all parts of the nervous system.
Point 2: The pH milieu in the central nervous system is constantly maintained at 7.4. The activity of ASICs channels could be rather low. The significance in the regulation by rNS of these channels in the neurons inside either central or peripheral nervous system remains to be resolved, though peripheral pain sensation could be strongly linked to these channels.
Response 2: We agree that the significance of ASICs regulation by nocistatin in neurons of the central and peripheral nervous systems needs to be resolved. However, the presence of additional targets on the neuron, in particular, unidentified to date G-coupled receptor(s), to which the peptide exhibits greater affinity, significantly complicates this task. It is also worth noting that the bi-directional effects of nocistatin demonstrated early in in vivo models can be partially explained via activating and inhibition of ASICs by nocistatin in our experiments.
Point 3: Lines 88-89, the osmolarity of ND-96 medium appears to be low (e.g., 90 mM NaCl). Would the difference between intracellular and extracellular osmolarity be possibly affected by nocistatin or pH? The ASICs-encoded inward currents shown in the study might have been overlapped with hypoosmolarity- or hyperosmolarity-induced inward currents.
Response 3: ND96 solution is a standard extracellular solution for working with oocytes (Virkki et al., JBC, 2002; Yang et al., J Vis Exp, 2011; Besson et al., Neuropharmacology, 2017; Dr. Kleinfeld Lab UC San Diego, Lab Manual for Oocyte Biophysics, 2014). The osmolarity of this solution, titrated with NaOH (value approximately 220 mOsmol), is comparable with intracellular osmolarity (varies from 275 to 325 mOsmol) (Gagné, Biochemical Ecotoxicology 2014, p. 21-31). The addition of the peptide even at a 1 mM concentration, as well as protons, by calculations (Koeppen & Stanton, Renal Physiology (Fifth Edition) 2013, p. 1-14), should not affect the difference between intracellular and extracellular osmolarity.
Point 4: All current traces shown in the manuscript appear to be unexpectedly clean. None of noisy appearances during the challenge of low pH or rNS were noted. Please compare the data shown in a previous paper (Figure 10A) by So et al. (Biochem Pharmacol 2018;151:79-88). For clarity, please quote this paper somewhere in the revised manuscript.
Response 4: We are grateful for helpful information about the activity of GMQ, and we included this in the manuscript (lines 43-44; ref.19).
Concerning the “purity” of the current traces presented in our manuscript, we would like to note that high magnitude of currents (several hundred nanoamperes to several microamperes) masks various noises and artifacts in traces, unlike the current trace presented in the paper by So et al. (Figure 10A) with, a few hundred picoamperes amplitude. Of course, in the manuscript we demonstrate the most successful records from many others we obtained. Some kind of noises and artifacts appearances during a solutions exchange can be found on Suppl. Figure 1 in the Supplementary Materials section, as well as low amplitude currents with noises that were recorded in this system earlier or in the article by Osmakov et al. (Front. Mol. Neurosci. 2017).
Point 5: The configuration and time course of current traces shown in the study are quite similar, though the values of standard errors (SEMs) in each parameter are high (Table 1). The holding currents in each trace are not changed (i.e., no leak components can be found out). Please describe the statement in detail in the Material and Methods section, regarding the experiments how the investigators determined whether the whole-cell recordings on oocytes were “unsuccessfully’ achieved.
Response 5: We added the statement “The whole-cell recording was performed if a response to the control pH stimulus was stable in at least two replicates.” (lines 100-101). We would also like to note that the data shown in Table 1 are the average of absolute values from different cells. Whereas the data shown in the Figures F-H are the result calculated from relative values, i.e. normalized data to the control of each cell. The normalization approach seems to us correct, since oocytes were not identical in the quantity and quality of the channels expressed, and, accordingly, responds to stimulus were different in amplitude. However, the highlighted above selection criterion was applied and the stability of cell response to the control pH stimulus in at least two replicates was measured.
Point 6: Line 91, please check the frequency value for filter. The value of 20 Hz seems to be so low.
Response 6: Such filtered frequency values are standardly used in the whole-cell voltage-clamp method with oocytes (for an example, please see also Joeres et al., Scientific Reports 2016; Reimers et al., PNAS 2017).
Point 7: Line 125, please change “rate” to “time course”. The data points of current decay were fitted.
Response 7: The change was done.
Point 8: Line 154-155, the sentence needs to be rephrased. “at a holding pH 7.4” is weird.
Response 8: The sentence was rephrased to “conditioning pH 7.4”.
Point 9: In lines 195-196, the sentence needs to be rephrased.to 1 mM nocistatin.
Response 9: The sentence was rephrased.
Point 10: Lines 158-159, to use “Each bar …” is more appropriate.
Response 10: The change was made.
Point 11: Whether rNS can perturb other types of ion channels such as epithelial Na+ channels still remains to be resolved.
Response 11: We agree that the epithelial sodium channels belonging to the same superfamily that the ASIC channels, as well as structurally close purinergic receptors, should be examined for whether nocistatin can affect them. We plan to do this in the future, but at this moment, unfortunately, we do not have such an opportunity.
Point 12: The holding potential used in these experiments was maintained at -50 mV. It is relatively hard to determine whether the reversal potential would have been changed in the presence of nocistatin.
Response 12: We believe that, since the mechanisms of ASIC channels opening by protons and nocistatin are similar (i.e., the kinetics of currents are similar), the ionic selectivity of the channels (or reversal potential) will not change upon activation by the peptide. However, we also plan to investigate this in the future.
Point 13: In Figure 2D, it seems difficult to realize why the values of Hill coefficient in dose-dependent relationship between different isoforms of ASICs channels are so large, since the size in the rNS peptide is extremely large. Additionally, please use large symbols for each data point in Figures 2C, 2D and 2E. Similarly, the x- and y-labeling in these panels should change to larger font size for clarity. Please also include the molecular weight of nocistatin in Materials and Methods section of the revised manuscript.
Response 13: Figure 2D shows the dose-response curve of ASIC1a channel activation by nocistatin (black line), and two inhibition curves of proton-induced current (named as 1st and 2nd components) of ASIC1a (blue and green lines, respectively). Last two curves are plotted against nocistatin concentration but for the current induced by following pH drop ( please, see Figure 2D legend). A high Hill coefficient values obtained as a result of any dose-dependence fitting means only a high level of cooperativity (Santillán, Mathematical Modelling of Natural Phenomena 2008). We believe that nocistatin can very likely act with a high degree of cooperativity on the channel consisting of three identical subunits.
We have increased the size of the symbols and the labels on the axes in Figures 2C, 2D and 2E. We also included the molecular weight of nocistatin in the text (line 75).
Point 14: The logisitic equations (lines 107-124) used in the manuscript appear to be complicated. For clarity, each formula should be amplified in the revised manuscript, if the investigators consider those as important to be used for analysis. If possible, please describe or define those logistic equations which have been previously used in the investigations on ASICs channels or other types of TRP channels in the Discussion section of the revised manuscript. Please include the relevant references into the revised manuscript.
Response 14: We gave a more detailed description of the equations F1, F2, and F3 (lines 113-119, 125-133). Equation F1 is usually used to quantify the activating (or potentiating) action of a ligand on a receptor (including ASIC channels) (Sherwood & Askwith, JBC 2008). However, it cannot give a satisfactory result for data fitting in some cases. The same phenomenon was observed for activation of ASIC1a by protons in several independent laboratories (Sherwood, JBC 2008; Stephan, Nature Communications 2018). Then we applied the equation F2, which is the result of the multiplication of two logistic equations and takes into account ligands independent binding in both the pool of high-cooperative sites and in one non-cooperative site (Osmakov et al., 2019). Fitting by this function yielded an excellent result, which once again confirms our assertion about similar mechanisms of ASIC channel activation by protons and the peptide. Equation F3 is commonly used to quantify the inhibitory effect of a ligand on a receptor (including ASIC channels) (Diochot, The EMBO Journal 2004; Reimers, PNAS 2017). We also included this description and the relevant references in the Results section (lines 211-213, 255-257).
Round 2
Reviewer 3 Report
The authors have answered most of questions concerned. However, to enhance the significant relevance with respect to this line of research, the statements regarding the responses to points 1, 2 and 4 are virtually important and hence need to be incorporated into the Discussion section of the revised manuscript.
Author Response
Point 1: The authors have answered most of questions concerned. However, to enhance the significant relevance with respect to this line of research, the statements regarding the responses to points 1, 2 and 4 are virtually important and hence need to be incorporated into the Discussion section of the revised manuscript.
Response 1: We added in the Discussion section the statements regarding the responses to points 1 (lines 302-311 of the revised manuscript), 2 (lines 320-322) and 4 (lines 277-278).